# AGENT²: AN AGENT-GENERATES-AGENT FRAMEWORK FOR REINFORCEMENT LEARNING AUTOMATION

## ABSTRACT

Reinforcement learning (RL) agent development traditionally requires substantial expertise and iterative effort, often leading to high failure rates and limited accessibility. This paper introduces Agent$^2$, an LLM-driven agent-generates-agent framework for fully automated RL agent design. Agent$^2$ autonomously translates natural language task descriptions and environment code into executable RL solutions without human intervention.

The framework adopts a dual-agent architecture: a Generator Agent that analyzes tasks and designs agents, and a Target Agent that is automatically generated and executed. To better support automation, RL development is decomposed into two stages—MDP modeling and algorithmic optimization—facilitating targeted and effective agent generation. Built on the Model Context Protocol, Agent$^2$ provides a unified framework for standardized agent creation across diverse environments and algorithms, incorporating adaptive training management and intelligent feedback analysis for continuous refinement.

Extensive experiments on benchmarks including MuJoCo, MetaDrive, MPE, and SMAC show that Agent$^2$ outperforms manually designed baselines across all tasks, achieving up to 55% performance improvement with consistent average gains. By enabling a closed-loop, end-to-end automation pipeline, this work advances a new paradigm in which agents can design and optimize other agents, underscoring the potential of agent-generates-agent systems for automated AI development.

## 1 INTRODUCTION

Reinforcement learning (RL) has achieved remarkable success in diverse domains such as robotics, games, and autonomous systems. However, the process of developing high-performing RL agents remains notoriously complex, requiring substantial domain expertise, intricate environment engineering, careful algorithm selection, and tedious trial-and-error tuning (Li, 2017). This manual, expertise-driven workflow has become a major barrier to the practical adoption and broader accessibility of RL, leading to high entry costs, limited scalability, and reduced research efficiency. Recent advances in large language models (LLMs) have demonstrated unprecedented capabilities in autonomous reasoning and code synthesis (Achiam et al., 2023). These advancements make it possible to automate not just component-level tasks, but the entire RL agent development pipeline without human intervention.

In this work, we present Agent$^2$, an agent-generates-agent framework that introduces a novel dual-agent architecture. The Generator Agent serves as an autonomous AI designer, capable of analyzing and producing all necessary components for an RL agent. Based on these components, the Target Agent is constructed and subsequently interacts with the environment for training and evaluation.

A key innovation of Agent$^2$ lies in its end-to-end automated pipeline for RL agent generation. Instead of merely automating isolated components, Agent$^2$ achieves unified and functionalized development of each module, efficient information exchange across modules, and integrated training frameworks. Moreover, Agent$^2$ incorporates mechanisms for detecting and resolving training anomalies, enabling iterative evolution and continual refinement. This comprehensive design trans-

forms RL development into a adaptive and progressively improving pipeline. The remainder of this paper is organized as follows. Section 2 reviews related work, Section 3 presents our methodology, Section 4 reports experimental results, and Section 5 concludes the paper.

## 2 RELATED WORK

Over the past few years, a wide range of general RL frameworks have been developed, such as RL-lib (Liang et al., 2018), Tianshou (Weng et al., 2022), and Xuance (Liu et al., 2023). These frameworks provide standardized implementations of algorithms, unified experiment management, and convenient interfaces for environment integration and distributed training. While these frameworks have greatly facilitated RL research and application, developing RL agents still heavily depends on expert knowledge and manual intervention. LLMs have enabled new opportunities for RL automation by intelligently generating or optimizing key components of the RL pipeline (Cao et al., 2024; Sun et al., 2024).

We contend that the RL automation can be broadly categorized into two complementary dimensions: MDP modeling and algorithmic optimization. For MDP modeling, some studies have explored automating the design of state representations, reward functions, and action spaces. LESR (Wang et al., 2024) and ExplorLLM (Ma et al., 2024a) use LLMs to autonomously generate or reformulate task-related state representations, improving learning efficiency and accelerating training. YOLO-MARL (Zhuang et al., 2024) and LERO (Wei et al., 2025) use LLMs to infer high-level or hidden context in multi-agent environment, helping to address limitations from partial observability. Some studies of reward function design have addressed the challenge of sparse rewards by developing automated reward shaping methods, especially for tasks that require complex robotic control (Ma et al., 2024b; Song et al., 2023) or operate in high-level environments such as Minecraft (Li et al., 2024). In multi-agent settings, research mainly focuses on solving the credit assignment problem for effective reward distribution (Nagpal et al., 2025; Lin et al., 2025; He et al., 2025). For action spaces, recent methods use LLM-generated action masking or suboptimal policies to dynamically constrain and guide RL agents, incorporating user preferences and improving sample efficiency and policy adaptability (Zhao et al., 2025; Karine & Marlin, 2025).

Research on its algorithmic optimization mainly follows the AutoML, which has seen rapid development and become relatively mature (He et al., 2021). Algorithmic optimization focusing on two directions: neural network architecture design and hyperparameter optimization. LLMatic (Nasir et al., 2024) combines LLM code generation with quality diversity optimization to discover diverse and effective architectures, while EvoPrompting (Chen et al., 2023) uses LMs as adaptive operators in evolutionary NAS. For hyperparameter optimization, some studies use LLMs to suggest and iteratively refine hyperparameter configurations based on dataset and model descriptions (Liu et al., 2024; Zhang et al., 2023; Mussi et al., 2023). Additionally, SLLMBO (Mahammadli & Ertekin, 2024) introduces a sequential LLM-based framework that adapts the search space dynamically and enhances optimization robustness. These methods highlight the potential of LLMs to design the network architecture and optimize the hyperparameter of RL algorithms.

However, most existing AutoRL approaches automate only a single stage of the RL pipeline, rather than enabling fully end-to-end agent generation, which requires the coordination of multiple modules as well as the handling of their interactions and debugging. This poses a far more challenging problem, which we address in the following section.

## 3 METHODS

This section provides an overview of the Agent$^2$ methodology, highlighting the automated workflow for RL agent generation. The overall process is illustrated in Figure 1. This section focuses on the Generator Agent, which serves as the core driver of automation in the Agent$^2$ framework. The Generator Agent is responsible for analyzing tasks, structuring RL knowledge, and autonomously generating all components required for agent construction.

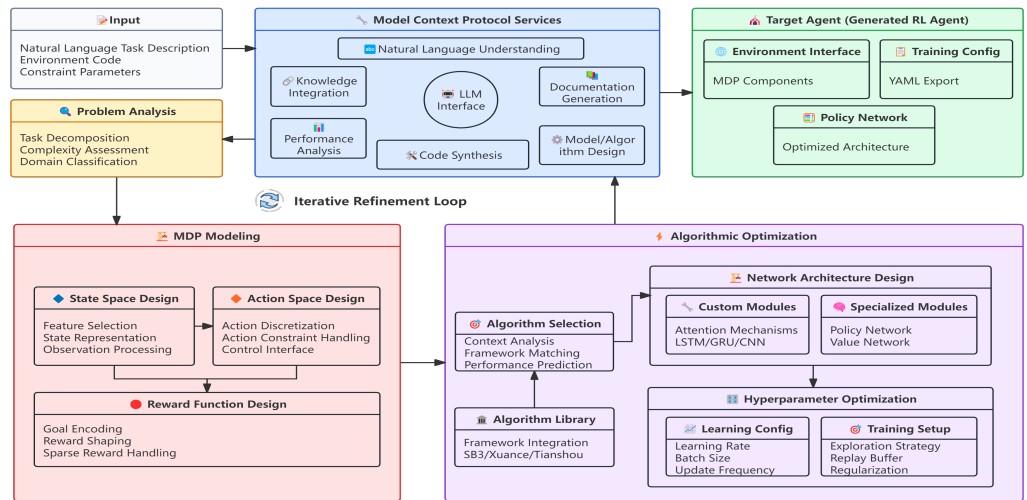

Figure 1: The framework of Agent$^2$ consists of three main stages. Firstly, Agent$^2$ analyzes the problem using natural language task descriptions and environment code as inputs. Secondly, the framework proceeds to MDP modeling, including the design of the state space, action space, and reward function. Thirdly, the framework is followed by algorithmic optimization, where the agent autonomously selects appropriate algorithms, designs network architectures, and tunes hyperparameters. The entire framework operates in compliance with the Model Context Protocol (MCP), ensuring standardized integration of services. Finally, the generated components are assembled into the Target Agent, which is ready for training and evaluation. The entire process supports iterative refinement to enhance solution quality.

### 3.1 TASK-TO-MDP MAPPING

#### 3.1.1 AUTOMATIC PROBLEM ANALYSIS

A key prerequisite for automated RL agent generation is the precise understanding and formalization of the target problem. The Generator Agent leverages LLMs to process heterogeneous user input with its knowledge bases, producing a structured representation of the RL problem. This analysis enables the system to: Identify learning objectives and reward structure from the task description; Infer state, action, and transition dynamics from the environment code; Extract explicit and implicit constraints, such as safety or operational limits; Anticipate challenges such as partial observability or sparse rewards.

Formally, let the user provide three natural language inputs: task description $T_{\text{task}}$, environment description or code $T_{\text{env}}$, and additional context $T_c$, such as constraints or prior knowledge. These are assembled into a prompt $P_{\text{analysis}}(T_{\text{task}}, T_{\text{env}}, T_c)$, which is processed by the LLM inference function $F_{\text{LLM}}$. The output $L_{\text{analysis}}$ is:

$$L_{\text{analysis}} = F_{\text{LLM}}(P_{\text{analysis}}(T_{\text{task}}, T_{\text{env}}, T_c)) \tag{1}$$

Here, $L_{\text{analysis}}$ is a structured analysis including formalized objectives, constraints, environment characteristics, and domain-specific challenges. This serves as the foundation for downstream MDP modeling and algorithm optimization, ensuring that all subsequent processes are both principled and context-aware.

#### 3.1.2 MDP MODELING

The MDP modeling module automates the refinement of initial RL environments into effective MDPs for reinforcement learning. In practice, available simulators or environments may only provide a coarse or incomplete MDP specification, requiring expert effort to redesign the observation space, action space, or reward function. This module leverages LLMs to analyze and optimize these core components, minimizing manual intervention. While LLMs are capable of constructing MDPs

from scratch, our approach focuses on the more practical and challenging scenario of automatically optimizing an existing but imperfect environment.

Formally, an MDP is defined as $\mathcal{M} = (\mathcal{S}, \mathcal{A}, \mathcal{P}, \mathcal{R}, \gamma)$, where $\mathcal{S}$ is the state (observation) space, $\mathcal{A}$ the action space, $\mathcal{P}$ the transition probability, $\mathcal{R}$ the reward function, and $\gamma$ the discount factor. Automated MDP modeling focuses on optimizing $\mathcal{S}$, $\mathcal{A}$, and $\mathcal{R}$, as these are central to agent-environment interaction.

The LLM examines environment code and Problem Analysis to identify and refine relevant state variables, possibly applying feature selection, dimensionality reduction, or combining observations to improve learning efficiency. The refined observation space is expressed as $s' = f_{\mathrm{obs}}(s) \in \mathcal{S}'$, where $f_{\mathrm{obs}}$ denotes the learned or synthesized transformation function.

Similarly, the action space is optimized to balance expressiveness and simplicity, with the LLM recommending merging, splitting, or reparameterization as needed: $a' = f_{\mathrm{act}}(a) \in \mathcal{A}'$, with $f_{\mathrm{act}}$ constructed through automatic analysis and synthesis.

For reward design, the LLM constructs functions that accurately reflect task objectives and promote effective learning, addressing issues such as sparsity or bias. The reward at time $t$ is $r_t = f_{\mathrm{rew}}(s_t, a_t, s_{t+1}, \mathrm{info})$, where $f_{\mathrm{rew}}$ is the synthesized reward function.

Each component is encapsulated as a wrapper function, enabling seamless integration and algorithmic verification within the RL framework.

### 3.1.3 Adaptive Verification and Refinement

Automatically synthesized MDP components may have executability issues or lead to suboptimal learning due to ambiguities in code generation. To address this, we introduce an adaptive verification and refinement framework that integrates generated components into the RL pipeline and iteratively improves them via automated validation and feedback.

Each component—observation, action, and reward functions ($f = \{f_{\mathrm{obs}}, f_{\mathrm{act}}, f_{\mathrm{rew}}\}$)—is first integrated through standardized interfaces and checked by a verification operator $V$. If verification fails, an error feedback $e$ is generated and, together with the original RL analysis $L_{\mathrm{analysis}}$ and component $f$, forms an adaptive prompt $P_{\mathrm{error}}$. The LLM inference function $F_{\mathrm{LLM}}$ then synthesizes a revised component $f^*$.

Beyond static verification, the system continuously monitors the agent's training dynamics and summarizes key performance indicators from TensorBoard data into a concise report, which serves as the performance feedback $\epsilon$ for further optimization. This feedback, together with $L_{\mathrm{analysis}}$, $f$, and training history $\mathcal{H}$, to construct a prompt $P_{\mathrm{perf}}$. The LLM then produces improved components. This closed-loop process ensures that MDP components are robust, reliable, and continually optimized, advancing the automation and practicality of LLM-based RL agent development.

The procedure is summarized in Algorithm 1.

### 3.2 Algorithmic Optimization

Upon completing the task-specific MDP design, Agent² automatically performs algorithm selection, network architecture design, training hyperparameter optimization, and integrated configuration with iterative refinement. The details of these four stages are presented in the following subsections.

### 3.2.1 Algorithm Selection

Algorithm selection aims to match the constructed MDP with an RL algorithm suitable for the environment's characteristics and task requirements. For example, value-based methods like DQN are appropriate for discrete action spaces, whereas policy gradient methods such as PPO and SAC are better for continuous control. Choosing the right algorithm is crucial for effective and stable policy learning.

Agent² uses LLMs to adaptively select algorithms based on comprehensive information, including the structured RL analysis $L_{\mathrm{analysis}}$, environment code $T_{\mathrm{env}}$, current MDP components $f =$

---

**Algorithm 1** Adaptive Verification and Refinement of MDP Components

---

**Require:** Initial MDP components $f^{(0)} = \{f_{\text{obs}}, f_{\text{act}}, f_{\text{rew}}\}$, LLM inference function $F_{\text{LLM}}$, RL analysis $L_{\text{analysis}}$, maximum iterations $T$
**Ensure:** Verified and optimized MDP components $f^*$
 1: Initialize best components $f^* \leftarrow f^{(0)}$, best score $S^*_{\text{hist}} \leftarrow S^{(0)}$, training history $\mathcal{H} \leftarrow \{\}$
 2: **for** $t = 1$ to $T$ **do**
 3:     Integrate $f^{(t)}$ into RL pipeline
 4:     Apply verification operator $V$ to $f^{(t)}$
 5:     **if** $V(f^{(t)}) = $ False with error $e$ **then**
 6:         $f^{(t+1)} \leftarrow F_{\text{LLM}}(P_{\text{error}}(e, L_{\text{analysis}}, f^{(t)}))$
 7:         **continue**
 8:     **end if**
 9:     Train agent with $f^{(t)}$, collect TensorBoard metrics $\epsilon^{(t)}$, obtain score $S^{(t)}$, update $\mathcal{H}$
10:     **if** $S^{(t)} > S^*_{\text{hist}}$ **then**
11:         $S^*_{\text{hist}} \leftarrow S^{(t)}, f^* \leftarrow f^{(t)}$
12:     **end if**
13:     **if** not converged **then**
14:         $f^{(t+1)} \leftarrow F_{\text{LLM}}(P_{\text{perf}}(\epsilon^{(t)}, L_{\text{analysis}}, f^{(t)}, \mathcal{H}))$
15:     **else**
16:         **break**
17:     **end if**
18: **end for**
19: **return** $f^*$

---

$\{f_{\text{obs}}, f_{\text{act}}, f_{\text{rew}}\}$, and training history $\mathcal{H}$. Given a set of candidates $G$, the LLM identifies the most suitable algorithm $g^* \in G$ and provides a brief rationale $L_{\text{algo}}$. Formally, the LLM inference function performs:

$$g^*, \ L_{\text{algo}} = F_{\text{LLM}}\big(P_{\text{alg}}(G, f, L_{\text{analysis}}, T_{\text{env}}, \mathcal{H})\big) \tag{2}$$

### 3.2.2 NETWORK ARCHITECTURE DESIGN

Effective reinforcement learning requires neural network architectures tailored to both the environment and the selected algorithm $g^*$. Agent[2] integrates information from observation and action space descriptions, the algorithm, structured reference architectures (e.g., $J_{\text{net}}$ in JSON), and statistical summaries from training history ($\mathcal{H}$). Using these structured prompts, the LLM critiques and refines previous designs, leveraging both domain expertise and empirical results to suggest architectural improvements.

Formally, the network design process is defined as:

$$N^*, \ L_{\text{net}} = F_{\text{LLM}}\Big(P_{\text{net}}(g^*, f_{\text{obs}}, f_{\text{act}}, J_{\text{net}}, \mathcal{H})\Big) \tag{3}$$

where $N^*$ is the recommended architecture and $L_{\text{net}}$ explains the design. This automated approach enables Agent[2] to generate adaptive network architectures for reliable RL training across diverse environments.

### 3.2.3 HYPERPARAMETER OPTIMIZATION

Training performance in RL depends heavily on careful hyperparameter selection and adjustment, such as learning rate, discount factor, batch size, and regularization. Agent[2] uses an LLM-driven process to systematically initialize and refine hyperparameters for the current environment, algorithm $g^*$, and network architecture $N^*$.

The Generator Agent consolidates structured information from the MDP model $f^*$, network $N^*$, selected algorithm $g^*$, reference configurations $J_{\text{hp}}$, and training history $\mathcal{H}$. Prompt engineering guides the LLM to assess current settings, identify bottlenecks, and propose incremental, interpretable adjustments prioritized for training stability and performance.

---

**Algorithm 2** Algorithmic Configuration Integration and Refinement

---

**Require:** Selected algorithm $g^*$, network architecture $N^*$, optimized hyperparameters $Hp^*$, optimized MDP components $f^*$, problem analysis $L_{\text{analysis}}$, LLM inference function $F_{\text{LLM}}$
**Ensure:** Final best configuration $\mathcal{C}^*$
1: Initialize best configuration $\mathcal{C}^* \leftarrow \mathcal{C}^{(0)}$, best score $S^*_{\text{hist}} \leftarrow S^{(0)}$, training history $\mathcal{H} \leftarrow \{\}$
2: **for** $t = 1$ to $T$ **do**
3:     Merge $g^*$, $N^*$, and $Hp^*$ into configuration $\mathcal{C}$
4:     Train agent with $\mathcal{C}^{(t)}$, collect TensorBoard metrics $\epsilon^{(t)}$, obtain score $S^{(t)}$, update $\mathcal{H}$
5:     **if** $S^{(t)} > S^*_{\text{hist}}$ **then**
6:         $S^*_{\text{hist}} \leftarrow S^{(t)}$, $\mathcal{C}^* \leftarrow \mathcal{C}^{(t)}$
7:     **end if**
8:     **if** not converged **then**
9:         Execute equations (2), (3), (4) to adaptive refinement $g^*$, $N^*$, $Hp^*$
10:     **else**
11:         **break**
12:     **end if**
13: **end for**
14: **return** Final best configuration $\mathcal{C}^*$

---

Formally, hyperparameter optimization is defined as:

$$Hp^*, \; L_{\text{hp}} = F_{\text{LLM}}\Big( P_{\text{hp}}(g^*, N^*, f^*, J_{\text{hp}}, \mathcal{H}) \Big) \tag{4}$$

where $Hp^*$ is the optimized hyperparameter set and $L_{\text{hp}}$ provides concise justifications for each change.

### 3.2.4 CONFIGURATION INTEGRATION AND REFINEMENT

After algorithm selection, network architecture design, and hyperparameter optimization, Agent² automatically integrates the selected algorithm $g^*$, network $N^*$, and optimized hyperparameters $Hp^*$ into a unified configuration $\mathcal{C}$, which is exported in standardized YAML format for compatibility and reproducibility. For adaptive refinement, Agent² monitors training performance via TensorBoard metrics and leverages the LLM to incrementally update $\mathcal{C}$ using observed results and historical data $\mathcal{H}$, forming a closed-loop process that enhances performance and robustness.

The pipeline of configuration integration and refinement is summarized in algorithm 2.

## 4 EXPERIMENTS

To comprehensively evaluate the effectiveness of Agent², we conduct experiments in progressive stages. We begin with a comparison under the MuJoCo benchmark, where our framework is evaluated against several well-established RL libraries, for which benchmark results are publicly available in this setting. We then extend the evaluation to a broader set of single-agent and multi-agent environments to test the generality and robustness of our approach. Finally, we perform ablation studies to quantify the respective impact of the two stages: Task-to-MDP mapping and algorithmic optimization.

### 4.1 EXPERIMENT SETUP

**Environments.** Single-agent environments include classic MuJoCo continuous control tasks—Ant, Humanoid, Hopper, and Walker2d. Each task requires an agent to control a simulated robot for stable and efficient locomotion. We also include the more challenging MetaDrive (Li et al., 2022), a large-scale autonomous driving simulator where agents must safely navigate diverse and dynamic traffic scenarios using rich sensory observations. Multi-agent environments include MPE (Lowe et al., 2017) with Simple Spread for cooperative landmark coverage and Simple Reference for communication-based target reaching, and SMAC (Samvelyan et al., 2019), which provides coop-

erative StarCraft II micromanagement tasks of varying scales and difficulties that require advanced coordination and tactical planning.

**Baselines.** We select widely used reinforcement learning algorithms as baselines to ensure the generality and credibility of our comparisons. For single-agent experiments, we employ PPO, SAC, and TD3 on the MuJoCo environments, and PPO and SAC on MetaDrive. For all multi-agent scenarios, we adopt MAPPO, a classic algorithm that has demonstrated strong performance in many benchmarks such as MPE and SMAC. As baselines, we use the original MDP model of all scenarios with the default hyperparameter settings of all algorithms, as implemented in the Xuance framework(Liu et al., 2023). In our framework, we employ Claude-Sonnet-3.7 as the LLM, which demonstrates stable performance in code generation. For more details about the experiment setup, please refer to Appendix A.

## 4.2 EXPERIMENTAL RESULTS

**MuJoCo benchmark comparison.** Firstly, we benchmark Agent$^2$ on a suite of MuJoCo tasks, directly comparing its performance with several widely used RL libraries. As shown in Table 1, since our framework is built on Xuance, we first compare directly against Xuance results. In almost all scenarios, Agent$^2$ outperforms Xuance, with substantial improvements in the Ant (e.g., PPO: 3831.9 vs.2810.7; TD3: 5981.4 vs.4822.9) and Humanoid (e.g., SAC: 6788.0 vs.4682.8) tasks; the only exception occurs in Hopper with PPO, where the performance is comparable (3444.6 vs.3450.1). Notably, the performance of Humanoid with TD3 in XuanCe benchmark is extremely poor (547.8) due to an unsuitable default configuration, while Agent$^2$ dramatically lifts the score to 5425.5. Compared to Tianshou, Agent$^2$ demonstrates clear advantages in most cases, with only two instances—Ant with SAC and Walker2d with SAC—where its performance is marginally weaker. Against SpinningUp, Agent$^2$ consistently outperforms across all reported benchmarks.

Table 1: The comparison between Agent$^2$ with classic RL benchmarks in Mujoco environment

| Scenario | Algorithm | XuanCe | Tianshou | SpinningUp | Agent$^2$ |
|----------|-----------|--------|----------|------------|-----------|
| Ant | PPO | 2810.7 | 3258.4 | 650 | **3831.9** |
| Ant | TD3 | 4822.9 | 5116.4 | 3800 | **5981.4** |
| Ant | SAC | 3499.7 | **5850.2** | 3980 | 4372.1 |
| Humanoid | PPO | 705.5 | 787.1 | - | **1085.9** |
| Humanoid | TD3 | 547.8 | 5189.5 | - | **5425.5** |
| Humanoid | SAC | 4682.8 | 5488.5 | - | **6788.0** |
| Hopper | PPO | **3450.1** | 2609.3 | 1850 | 3444.6 |
| Hopper | TD3 | 3492.4 | 3472.2 | 3564.1 | **3570.1** |
| Hopper | SAC | 3517.4 | 3542.2 | 3150 | **3576.5** |
| Walker2d | PPO | 4318.6 | 3588.5 | 1230 | **4649.5** |
| Walker2d | TD3 | 4307.9 | 3982.4 | 4000 | **4844.4** |
| Walker2d | SAC | 4730.5 | **5007.0** | 4250 | 4805.3 |

**Generalization to broader environments.** To assess generality and robustness beyond MuJoCo, we further evaluate Agent$^2$ across a wider range of single-agent and multi-agent environments. Since these extended environments lack publicly available benchmark results, we compare Agent$^2$ against the Xuance framework, ensuring fairness by adopting the same training budget and reporting the best performance achieved.

In Ant scenario of MuJoCo environment, the training curves (see Fig. 2a, 2b, 2c) show substantial improvements in sample efficiency and final performance, especially for TD3 and SAC. After the baseline quickly converges, the episode rewards of Agent$^2$ continue to increase significantly in the later stages of training. In the more challenging Humanoid environment, The corresponding training curves (see Fig. 2d, 2e, 2f) illustrate: while Agent$^2$ with PPO shows a modest improvement after the baseline stabilizes, Agent$^2$ with SAC and TD3 display large further gains after the baseline converges, highlighting the strong late-stage optimization of Agent$^2$.

On the MetaDrive autonomous driving environment, Agent$^2$ consistently outperforms the baseline for both PPO and SAC. The training curves in Fig. 3a, 3b show that, Agent$^2$ with PPO achieves

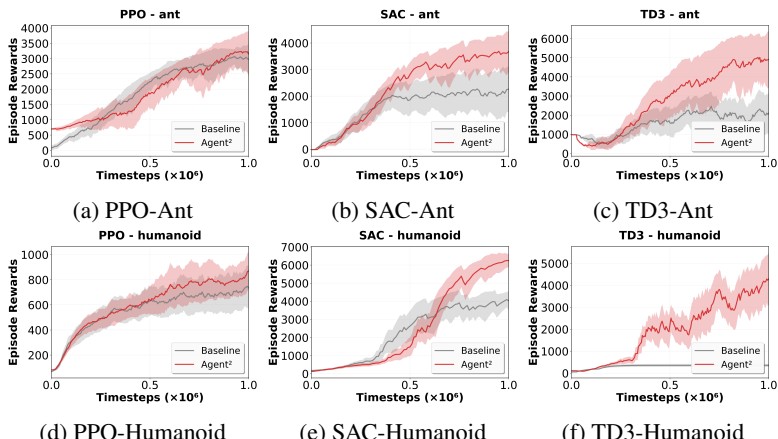

Figure 2: Training curve comparisons on MuJoCo environments.

a moderate increase in the final convergence level, while Agent$^2$ with SAC substantially accelerates early training progress and rapidly surpasses the baseline from the beginning. These results highlight the effectiveness of automated optimization even in complex, real-world-like single-agent environments.

On MPE cooperative tasks, Agent$^2$ brings modest yet consistent gains in team performance. In Simple Spread, the average episode reward improves from -19.73 to -16.31, while in Simple Reference, it increases from -19.61 to -19.00. The corresponding training curves (Fig. 3c, 3d) show these steady improvements over the baseline.

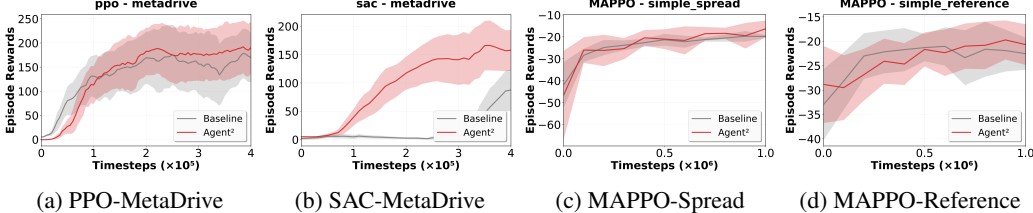

Figure 3: Training curve comparisons on MetaDrive and MPE environments.

On challenging SMAC scenarios, Agent$^2$ consistently improves over the baseline. The training curves in Fig.4a, 4b, 4c show that in the 8m scenario, Agent$^2$ achieves both faster convergence and substantially higher final episode rewards compared to the baseline. In the more difficult 1c3s5z scenario, Agent$^2$ initially matches the baseline, but gradually surpasses it in cumulative rewards and final performance as training progresses. The win rate curves (Fig.4d, 4e, 4f) reflect similar trends, with win rates rising from 0.77 to 0.94 in 8m, from 0.82 to 0.85 in 2s3z, and from 0.17 to 0.23 in 1c3s5z. Overall, the results indicate that Agent$^2$ delivers consistent performance improvements, particularly in complex scenarios, thereby validating the effectiveness of automated optimization for multi-agent tactical cooperation.

**Ablation Analysis.** To rigorously demonstrate the effectiveness of our two-stage pipeline, we quantitatively compare the baseline (default manual configuration), after Task-to-MDP Mapping (Stage 1), and after subsequent Algorithmic Optimization (Stage 2). As summarized in Figure 5, most environments and algorithms show clear improvements: Task-to-MDP Mapping improves performance in 83% of scenarios, and Algorithmic Optimization delivers additional gains in 67% of cases. We exclude the humanoid-TD3 experiment, as it exhibited abnormal behavior in the benchmark comparison.

Compared with the baseline, Stage 1 optimizes the task formulation by reconstructing the MDP, delivering substantial gains across domains. For example, in Ant-TD3 the reward increases by 55% (3853.8 → 5981.4), in MetaDrive-SAC by 23% (178.2 → 219.6), and in the challenging SMAC-

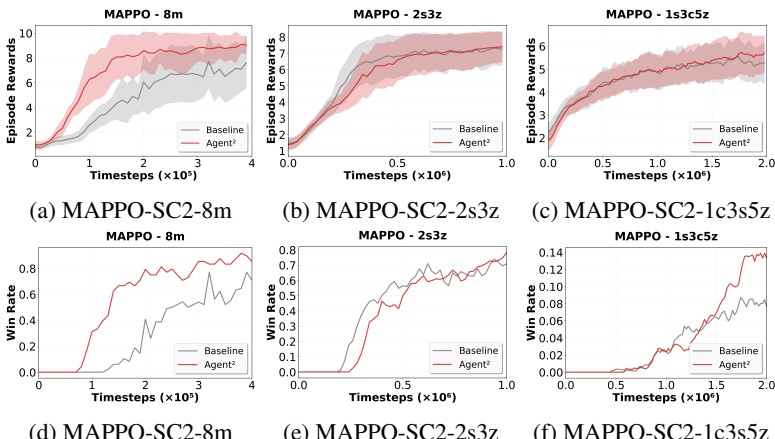

(a) MAPPO-SC2-8m  (b) MAPPO-SC2-2s3z  (c) MAPPO-SC2-1c3s5z

(d) MAPPO-SC2-8m  (e) MAPPO-SC2-2s3z  (f) MAPPO-SC2-1c3s5z

Figure 4: Training curve comparisons on StarCraft-II environments.

1c3s5z scenario, the win rate improves from 0.17 to 0.21 (a 25% relative gain). Building upon this, Stage 2 further enhances performance by refining algorithmic configurations: Ant-PPO achieves an additional 13% improvement (3385.4 → 3831.9), Humanoid-SAC rises by 11.6% (6081.2 → 6787.9), MetaDrive-SAC gains another 18% (219.6 → 259.8), and SMAC-1c3s5z advances from 0.21 to 0.23 (a further 10% relative gain). Together, these stages enable robust and fully automated generation of high-performing RL agents across a diverse range of single-agent and multi-agent benchmarks. Detailed statistics can be found in Appendix B.

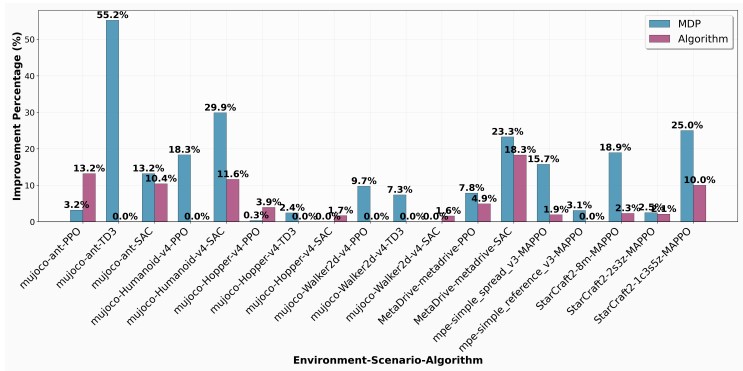

Figure 5: Performance improvement across Task-to-MDP Mapping and Algorithmic Optimization.

## 5 CONCLUSION

In this work, we present Agent[2], a fully automated, LLM-driven agent-generates-agent framework that advances the automation, accessibility, and performance of RL development. By introducing a novel dual-agent architecture and two-stage pipeline separating MDP modeling from algorithmic optimization, Agent[2] autonomously translates high-level task descriptions and raw environment code into high-performing RL agents without human intervention.

Extensive experiments on single-agent and multi-agent benchmarks show that Agent[2] outperforms the majority of manually designed baselines across diverse environments and algorithms, demonstrating both the effectiveness and generality of the approach. Ablation analysis further highlights the complementary roles of automated environment adaptation and iterative algorithmic refinement: Agent[2] boosts performance in 83% of task-algorithm pairs after MDP adaptation, with further gains in 67% of cases following optimization. Together, these findings highlight the potential of Agent[2] for real-world deployment and point toward future extensions to increasingly complex tasks and domains.

## ETHICS STATEMENT

This work complies with the ICLR Code of Ethics. Our research does not involve human subjects, personally identifiable information, or sensitive data. All experiments are conducted on widely used, publicly available reinforcement learning benchmarks (e.g., MuJoCo, MetaDrive, MPE, SMAC), which are appropriately credited. The proposed framework focuses on automating RL agent development within controlled simulation environments, and we identify no foreseeable risks related to privacy, security, or social fairness. We are committed to maintaining research integrity, transparency, and reproducibility throughout this study.

## REPRODUCIBILITY STATEMENT

We recognize that employing large language models (LLMs) in our framework introduces inherent stochasticity, which may prevent exact replication of every generated component. To mitigate this limitation, we provide extensive implementation details, including a thorough description of the overall framework, the prompt templates used at each stage, the number of iterative refinement rounds, and the specific LLM configurations (model identifiers, temperature settings, and other decoding parameters). Furthermore, all empirical evaluations are conducted on widely used and publicly accessible benchmarks, for which datasets and environments are available through their official repositories. Together, these resources are intended to enable independent researchers to approximate our results, verify the key findings, and build upon this work with confidence.

## USE OF LARGE LANGUAGE MODELS

In preparing this paper, we employed large language models (LLMs) exclusively as a writing assistance tool. Specifically, LLMs were used to help polish grammar, improve clarity during the manuscript preparation. All research ideas, problem formulations, experiments, analyses, and conclusions were conceived and executed entirely by the authors. The LLM did not contribute to research ideation, design of methods, implementation, or data analysis.

The authors take full responsibility for all content presented in this work, including sections where LLMs were used for language refinement. LLMs were not involved in generating scientific claims or results, and are not considered contributors or authors of this paper.

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

## A    EXPERIMENTAL DETAILS.

### A.1    ENVIRONMENTS.

For the **MuJoCo** suite, we consider four classic continuous control tasks. In **Ant**, the agent controls a four-legged ant robot and aims to move forward as fast and smoothly as possible, keeping the body stable and preventing falls. The **Humanoid** scenario is more challenging, requiring the agent to coordinate a high-dimensional humanoid robot to walk quickly and stably while maintaining balance. In **Walker2d**, the agent must control a two-legged bipedal robot to walk forward, applying torques to six joints to ensure stability. The **Hopper** task involves a single-legged robot that must hop forward rapidly and avoid falling, requiring fine control over three actuated joints.

**MetaDrive** serves as a large-scale autonomous driving simulator, where the agent must safely drive a vehicle through randomly generated roads and traffic scenarios. The objective is to reach the destination efficiently while avoiding collisions with other vehicles and obstacles. The observation space incorporates ego-vehicle states, navigation cues, and lidar-like sensor data, providing a comprehensive view of the driving environment.

The **Multi-Agent Particle Environment (MPE)** is used to evaluate multi-agent cooperation and communication. In the **Simple Spread** scenario, multiple agents must collaborate to cover all target landmarks on the map while minimizing collisions with each other. Each agent observes its own velocity and position, as well as information about landmarks and other agents. In the **Simple Reference** scenario, two agents are each tasked with approaching specific colored landmarks, but the assignment is communicated by the other agent. Agents must leverage both movement and communication actions to succeed, receiving both individual and global rewards.

For more complex cooperative tasks, we use the **StarCraft Multi-Agent Challenge (SMAC)** benchmark. In **8m**, eight allied Marines must coordinate to defeat eight enemy Marines on a symmetric map with no obstacles, emphasizing position and focus fire. The **2s3z** scenario is heterogeneous, featuring teams of two Stalkers and three Zealots each, which requires the agent to adapt strategies for mixed unit types. **1c3s5z** is a large-scale heterogeneous battle with one Colossus, three Stalkers, and five Zealots per side, demanding advanced coordination and tactical planning due to unit diversity and increased team size.

### A.2    EXPERIMENTAL PARAMETERS

In our experiments, we conducted a thorough evaluation of the proposed framework across multiple environments and algorithms. All experiments are conducted on a workstation equipped with a 22-core AMD EPYC 7402 CPU, a NVIDIA RTX 4090 GPU, and 108 GB of RAM. Throughout the automated process, we employ Claude-Sonnet-3.7 as the underlying LLM. This model has shown stable performance in both problem analysis and code generation, and thus serves as a dependable backbone for our framework. Table 2 provides a summary of the model parameters used in our experiments.

Table 2: Model inference parameters.

| Parameter | Value | Description |
|---|---|---|
| Context length | 128K | Maximum input tokens supported by the model |
| Max_tokens | 1024 | Maximum tokens generated in one response |
| Temperature | 0.6 | Controls randomness; lower = more deterministic |
| Top_p | 0.7 | Nucleus sampling cutoff; balances diversity |
| Top_k | 50 | Limits sampling to the top-$k$ probable tokens |

To ensure reproducibility and provide a clear understanding of our experimental setup, we established standardized parameters for all evaluations. In terms of training settings, we carefully balanced training costs with performance outcomes to configure appropriate parameters that would demonstrate the effectiveness of our approach across diverse scenarios without excessive computational requirements. Table 3 presents the experimental configurations of each environment, including the training steps, the evaluation episodes, and the number of automated optimization iterations.

Table 3: Experimental settings for different environments

| Environment | Training Steps | Evaluation episodes | iterations per stage |
|---|---|---|---|
| MuJoCo | 1M | 50 | 5 |
| MetaDrive | 400K | 50 | 5 |
| MPE | 1M | 50 | 5 |
| SMAC-8m | 400K | 50 | 3 |
| SMAC-2s3z | 1M | 50 | 3 |
| SMAC-1c3s5z | 2M | 50 | 3 |

## A.3 LLM PROMPT DESIGN

To clearly describe the design workflow, we provide the complete LLM prompt used in our study as follows.

```
<System>
You are a professional reinforcement learning problem analysis expert.
Please analyze the given problem and output the analysis results in the
    following format:

# Task Objectives
[Describe specific optimization goals]

# Constraints
[List main limitations]

# Environment Characteristics
- Deterministic/Stochastic: [Explanation]
- Fully/Partially Observable: [Explanation]
- Single-agent/Multi-agent: [Explanation]

# Key Challenges
[List 2-3 main challenges]

<User>
Please analyze the following reinforcement learning
    problem:\n{problem_description}
Environment code:\n{env_code}
```

Listing 1: Problem Analyzing Prompt

```
<System>
You are an expert RL observation space designer specializing in
    optimizing state representations.
Your task is to design the most effective observation space based on
    problem description and analysis, and implement an ObsWrapper
    function to transform original observation vectors into new
    observation vectors.

Design Objectives:
1. Improve agent learning efficiency through better state representation
2. Enhance feature extraction and information utilization
3. Normalize and scale values appropriately
4. Balance information richness with dimensionality

Pay attention to division by zero issues in floating-point scalar
    operations, avoid generating nan in calculations.

Please output strictly in the following format:
---
```

```
# Observation Variables
[List main observation variables and their meanings]

# Observation Space Definition of Single-agent
```json
{"dim": dimension, "type": "continuous/discrete", "low": minimum_value,
    "high": maximum_value}
```

# Observation Space Definition of Multi-agent
```json
{
    "agent_0": {"dim": dimension, "type": "continuous/discrete", "low":
        minimum_value, "high": maximum_value},
    "agent_1": {"dim": dimension, "type": "continuous/discrete", "low":
        minimum_value, "high": maximum_value},
    ...
}
```

# ObsWrapper Implementation of Single-agent
```python
def custom_state_transform(state: np.ndarray) -> np.ndarray:
    # Implement state transformation logic
    return transformed_state
```

# ObsWrapper Implementation of Multi-agent
```python
def custom_state_transform(state: Dict[str, np.ndarray]) -> Dict[str,
    np.ndarray]:
    # Transform each agent separately
    transformed_state = {}
    for agent_id, agent_obs in state.items():
        # Transformation logic
        transformed_state[agent_id] = ...
    return transformed_state
```

# Design Description
[Brief explanation of design ideas and considerations, including:
1. Key transformations applied and their purpose
2. How this design addresses the specific challenges of the environment
3. Expected impact on agent learning]

<User>
Task Environment: {env_name} - {scenario_name}
Problem Description: {problem_description}
Problem Analysis: {analysis_str}
Environment Code: {env_code}
default Observation Space: {default_observation_space}

Design Requirements:
1. Focus on extracting meaningful features from raw observations
2. Apply appropriate normalization to stabilize learning
3. Consider feature engineering that highlights task-relevant information
4. Maintain numerical stability in all transformations
5. If the observation space is discrete, the space dim equals the number
    of discrete observations
```

Listing 2: Observation Space Designing Prompt

```
<System>
```

```
You are an expert RL action space designer specializing in optimizing
    action representations.
Your task is to design the most effective action space based on problem
    description, analysis and observation space,
and implement an ActionWrapper function that maps the designed action
    vector to the environment's original action space.

Design Objectives:
1. Improve agent learning efficiency through better action representation
2. Create intuitive action mappings that facilitate policy learning
3. Balance expressiveness with simplicity
4. Ensure smooth transitions between action spaces

Pay attention to division by zero issues in floating-point scalar
    operations, avoid generating nan in calculations.
Please output strictly in the following format:

# Action Variables
[List main action variables and their meanings]

# Action Space Definition of Single_agent
```json
{"dim": dimension, "type": "continuous/discrete", "low": minimum_value,
    "high": maximum_value}
```

# Action Space Definition of Multi-agent
```json
{
    "agent_0": {"dim": dimension, "type": "continuous/discrete", "low":
        minimum_value, "high": maximum_value},
    "agent_1": {"dim": dimension, "type": "continuous/discrete", "low":
        minimum_value, "high": maximum_value},
    ...
}
```

# ActionWrapper Implementation of Single_agent
```python
def custom_action_transform(custom_action: np.ndarray) -> np.ndarray:
    # Implement action transformation logic
    return gym_action
```

# ActionWrapper Implementation of Multi-agent
```python
def custom_action_transform(custom_action: Dict[str, np.ndarray]) ->
    Dict[str, np.ndarray]:
    # Transform each agent separately
    gym_action = {}
    for agent_id, agent_action in custom_action.items():
        # Transformation logic
        gym_action[agent_id] = ...
    return gym_action
```

# Design Description
[Brief explanation of design ideas and considerations, including:
1. Key transformations applied and their purpose
2. How this design addresses the specific challenges of the environment
3. Expected impact on agent learning]

<User>
Task Environment: {env_name} - {scenario_name}
Problem Description: \n{problem_description}
```

```
Problem Analysis: \n{analysis_str}
Environment Code: \n{env_code}
default Action Space: \n{default_action_space}
Implemented Observation Transformation Code: \n{obs_code}
Implemented Observation Space: \n{obs_space}

Design Requirements:
1. Compatibility with transformed observation space features
2. Smoothness and numerical stability of action transformation
3. Avoid action space being too complex leading to training difficulties
4. Create intuitive mappings between designed actions and environment
   actions
5. If the action space is discrete, the space dim equals the number of
   discrete actions
```

Listing 3: Action Space Designing Prompt

```
<System>
You are an expert RL reward function designer specializing in optimizing
    reward signals.
Your task is to design the most effective reward function based on
    problem description, analysis, environment code, observation space,
    and action space,
and implement a RewardWrapper function that calculates rewards based on
    custom current state, action, next state and environment information.

Design Objectives:
1. Create a reward signal that effectively guides learning toward
   desired behaviors
2. Balance immediate feedback with long-term goals
3. Avoid reward hacking and degenerate policies
4. Ensure numerical stability and proper scaling

Pay attention to division by zero issues in floating-point scalar
    operations, avoid generating nan in calculations.
Please output in the following format:

# Reward Calculation Logic
[Explain the main components and weights of reward calculation]

# RewardWrapper Implementation of Single_agent
```python
def custom_reward_function(
    custom_current_state: np.ndarray,
    custom_action: np.ndarray,
    custom_next_state: np.ndarray,
    info: dict
) -> float:
    # Implement reward calculation logic
    return custom_reward
```

# RewardWrapper Implementation of Multi-agent
```python
def custom_reward_function(
    custom_current_state: Dict[str, np.ndarray],
    custom_action: Dict[str, np.ndarray],
    custom_next_state: Dict[str, np.ndarray],
    info: dict
) -> Dict[str, float]:
    # Calculate rewards for each agent separately
    custom_reward = {}
    for agent_id in custom_current_state:
        # Reward calculation logic
```

```
        custom_reward[agent_id] = ...
    return custom_reward
```

# Design Description
[Brief explanation of design ideas and considerations, including:
1. Key reward components and their purpose
2. How this design addresses the specific challenges of the environment
3. Expected impact on agent learning]

<User>
Task Environment: {env_name} - {scenario_name}
Problem Description: {problem_description}
Problem Analysis: {analysis_str}
Environment Code: {env_code}
Designed Observation Space Code: {obs_code}
Implemented Observation Space: {obs_space}
Designed Action Space Code: {action_code}
Implemented Action Space: {action_space}

Design Requirements:
1. Input states have been transformed by ObsWrapper, please refer to
    ObsWrapper code to understand state structure
2. Prioritize using info dictionary to get environment information,
    avoid hard-coding extraction from state vectors
3. Ensure numerical stability of reward calculation, avoid nan or inf
    values
4. Create a reward structure that guides learning toward optimal behavior
5. Balance immediate feedback with long-term goals
```

Listing 4: Reward function Designing Prompt

```
<System>
You are an expert in reinforcement learning neural network architecture
    design.
Given the observation space and action space, please design the most
    suitable network architecture for the problem.
For each item, strictly choose only from the options listed in
    parentheses.

Your response should follow this format:
# Network Architecture and Parameters, provide detailed parameter
    configuration, including:
1. Layer Types and Dimensions (choose only from: Basic_MLP,
    Basic_Identical, Basic_CNN, Basic_RNN; specify dimensions)
2. Activation Functions (choose only from: relu, leaky_relu, tanh,
    sigmoid, softmax, elu)
3. Regularization Methods (choose only from: LayerNorm, BatchNorm,
    BatchNorm2d)
4. Other Special Configurations (if any; otherwise leave blank)

# Design Description
[Brief explanation of design ideas and considerations]

<User>
Task Environment: {env_name} - {scenario_name}
Algorithm: {algorithm}
Target Parameters for Optimization: {list(network_config.keys())}
Observation Space: {obs_design_str}
Action Space: {action_design_str}
```

Listing 5: Network Designing Prompt

```
<System>
You are an expert RL hyperparameter tuner specializing in optimizing
    specific training parameters.
Your job is to set appropriate training hyperparameters based on
    environment, algorithm, neural network design and reference
    configuration.

Tuning Objectives:
1. Improve the overall performance.
2. Enhance training stability.
3. Respect computational constraints.
4. Favor incremental improvements where possible.

Recommended Hyperparameter Tuning Priorities:
Learning Rate: A crucial factor for training stability and convergence.
Discount Factor (gamma): Balances the importance of immediate and future
    rewards. Common values range from 0.97 to 0.995.
GAE Lambda (gae_lambda): Adjusts the bias-variance trade-off in
    advantage estimation. Typical values are between 0.92 and 0.97.
Clip Range: Helps maintain stable policy updates. Consider values within
    [0.1, 0.3].
Batch Size and Update Frequency: Parameters such as horizon size, number
    of epochs, and minibatch count should be tuned to balance learning
    speed and stability.
Value Function and Entropy Coefficients: Tuning the value loss
    coefficient (e.g., 0.1-0.5) and entropy coefficient (e.g., 0.0-0.01)
    can improve value estimation and exploration, respectively.
Gradient Clipping: Prevents large, destabilizing updates. Experiment
    with different gradient clipping norms, such as 0.5 or 1.0.
Observation and Reward Normalization: Enabling and tuning normalization
    can enhance training stability and generalization.

Output Format:
```yaml
parameter_name: value
# Reason: [problem identified] - [why change helps] - [expected outcome]
```

Requirements:
- Start hyperparameter searches with the learning rate and clip range,
    then proceed to gamma, gae_lambda, and batch/update parameters.
    Fine-tune normalization and coefficient settings in later stages for
    best results.
- Suggest changes to only relevant parameters (max 5).
- Use incremental changes (less than 30%) unless fixing critical
    stability issues.
- Ensure values remain within safe, specified ranges.
- Consider parameter interactions and algorithm-specific constraints.
- Give concise, clear reasoning for each suggestion.

<User>
Task Environment: {env_name} - {scenario_name}
Algorithm: {algorithm}
Network Architecture: {network_design}
Target Parameters for Optimization: {list(algorithm_config.keys())}
Observation Space: {obs_design_str}
Action Space: {action_design_str}
Please analyze the current configuration and determine if any of the
    target parameters need optimization.
```

Listing 6: Hyperparameters Designing Prompt

# B EXPERIMENTAL RESULTS

To demonstrate the effectiveness of our two-stage pipeline, we quantitatively compare the baseline (default manual configuration), after Task-to-MDP Mapping (Stage 1), and after subsequent Algorithmic Optimization (Stage 2). The detailed results are summarized in Table 4.

Table 4: Performance comparison on 2 stages

| Environment | Scenario | Algorithm | Baseline | Stage1 | Stage2 |
|---|---|---|---|---|---|
| MuJoCo | Ant | PPO | 3280.51 | 3385.38 | 3831.86 |
| MuJoCo | Ant | TD3 | 3853.84 | 5981.39 | 5981.39 |
| MuJoCo | Ant | SAC | 3499.76 | 3960.73 | 4372.05 |
| MuJoCo | Humanoid | PPO | 917.64 | 1085.85 | 1085.85 |
| MuJoCo | Humanoid | TD3 | 354.79 | 5425.46 | 5425.46 |
| MuJoCo | Humanoid | SAC | 4682.83 | 6081.16 | 6787.94 |
| MuJoCo | Hopper | PPO | 3304.03 | 3314.88 | 3444.56 |
| MuJoCo | Hopper | TD3 | 3486.29 | 3570.13 | 3570.13 |
| MuJoCo | Hopper | SAC | 3517.41 | 3517.41 | 3576.54 |
| MuJoCo | Walker2d | PPO | 4236.68 | 4649.51 | 4649.51 |
| MuJoCo | Walker2d | TD3 | 4513.16 | 4844.38 | 4844.38 |
| MuJoCo | Walker2d | SAC | 4730.53 | 4730.53 | 4805.28 |
| MetaDrive | metadrive | PPO | 245.34 | 264.57 | 277.54 |
| MetaDrive | metadrive | SAC | 178.16 | 219.60 | 259.76 |
| MPE | simple_spread_v3 | MAPPO | -19.73 | -16.63 | -16.31 |
| MPE | simple_reference_v3 | MAPPO | -19.61 | -19.00 | -19.00 |
| SC2 | 8m | MAPPO | 0.77 | 0.92 | 0.94 |
| SC2 | 2s3z | MAPPO | 0.82 | 0.84 | 0.85 |
| SC2 | 1c3s5z | MAPPO | 0.17 | 0.21 | 0.23 |

