# OpenReview forum: "Agent²: An Agent-Generates-Agent Framework for Reinforcement Learning Automation"
_ICLR.cc/2026/Conference — Submitted to ICLR 2026_

### Official Review · Reviewer_HBay · 2025-10-24

**Soundness:** 3
**Presentation:** 3
**Contribution:** 3
**Rating:** 4
**Confidence:** 3

**Summary:**

The authors present Agent^2, an LLM-based framework for creating MDPs and agent architectures (including hyperparameters) to automate the reinforcement learning agent development pipeline. The authors compare the results of their method (i.e. the performance of the Target Agents generated by Agent^2) to the baseline benchmark agents for various environments, including those in MuJoCo, MetaDrive, MPE, and SMAC.

1. **What is the specific question and/or problem tackled by the paper?**

    The problem of automating RL agent design.

2. **Is the approach well motivated, including being well-placed in the literature?**

    It appears that the work is well-placed in the literature.

3. **Does the paper support the claims? This includes determining if results, whether theoretical or empirical, are correct and if they are scientifically rigorous.**

    The Target Agent performance beats the baselines in the vast majority of environments.

4. **What is the significance of the work? Does it contribute new knowledge and sufficient value to the community?**

    The paper contributes significant value by virtue of automating the design of RL agents.

**Strengths:**

The paper is structured and written in a logical way, the algorithms are generally clear and possible to follow, the results are interpretable and clear, and the originality is there.

**Weaknesses:**

The paper lacks some examples of some of the outputs generated by the LLMs, e.g. L_algo and L_analysis, which would aid in understanding what the language agents are doing. Overall the paper is presenting a huge framework. Supplemental materials should be submitted and reviewed.

**Questions:**

Examples of L_analysis, L_algo, etc. for an example environment would be useful to include in the appendix.

Analysis of the Task-To-MDP Mapping would be useful and interesting.

Supplementary material would be useful. Where is the codebase? Rejecting until the codebase can be verified.

---

### Official Review · Reviewer_7D2m · 2025-10-27

**Soundness:** 3
**Presentation:** 3
**Contribution:** 2
**Rating:** 2
**Confidence:** 4

**Summary:**

The authors provide a framework for automatically designing the formalization of an RL problem and selecting the most suitable algorithm to solve it. They make use of an LLM approach, where the user is required to specify the problem, and the final outcome is a formalization (via code) of the (problem, algorithm) pair. They tested their approach vs. benchmarks present in the literature and showed that it is providing a better solution than the ones using standard parameter configurations

**Strengths:**

- The framework is clearly presented and described
- The work goes in the direction of the democratization of RL

**Weaknesses:**

- The English should be polished
- If I understood correctly, the final outcome is a set of properly designed prompts, which does not seem like a rigorous methodology
- The level of the user and the  knowledge required to describe the problem is not discussed or assessed
- The pseudocode is not explained properly
- No considerations about the difference in terms of modeling between the proposed solution and the benchmark have been provided
- No clear information about the explanation of the models selected is provided
- No code or tool is provided or claimed will be distributed

**Questions:**

1) In the MDP modeling, how can one determine the best features for the problem at this stage?
2) What is info in f_rew?
3) It is not clear what the verification operator is doing. Even if the meaning is evident, how it would verify that a model is appropriate and functioning is not clear from your description. The pseudocode is not helping the understanding
4) It is not clear to me if the LLM component is using a general-purpose LLM or models that you trained or fine-tuned over the specific RL setting.
5) Is it really necessary to do hyperparameter tuning with an LLM? You should motivate why it is better than using standard optimization procedures.
6) Algorithm 2: The pseudocode you provided is too generic. You should describe it more specifically, especially when specifying the termination conditions.
7) Are the results in Table 1 the result of a single run learning procedure, or is the uncertainty about them available somehow?
8) What are you going to release as a tool to be used by the RL community?

---

### Official Review · Reviewer_R2dm · 2025-10-30

**Soundness:** 2
**Presentation:** 3
**Contribution:** 3
**Rating:** 4
**Confidence:** 3

**Summary:**

The paper presents Agent². It is a framework that uses an LLM to design and train the RL agent for certain tasks. The LLM analyzes the environment, defines the MDP elements (observations, actions, rewards) and picks an algorithm and hyperparameters to train the agent. Experiments on several benchmarks show consistent improvements.

**Strengths:**

The two-step procedure of MDP design and algorithmic optimization is reasonable. The writing is easy to understand.

The evaluation is convincing. Both single and multi-agent domains are covered. Strong baselines are compared with. The results support that the proposed approach outperforms the baselines most of the time. Ablation studies are also included.

**Weaknesses:**

The paper is lacking many training details in general. The appendix only provides the prompts and many details remain unclear. For instance, what do all the Layers specifically mean in Listing 5 and how are they converted into a real network? How large are they? Given only the current version of the paper, the reproducibility is limited.

As we know the LLMs are fairly random, it is important to have multiple runs of the whole framework (call LLM multiple times) and report the mean and standard deviation in table 1. Currently it only reports a single run and is thus less convincing. This point also goes back to the reproducibility.

I am not sure if the authors have taken the computation on the LLM calls into account when comparing with the baselines. The LLM calls spend extra compute and wall time.

**Questions:**

What happens if the LLM produces faulty output? For example, if the generation is not executable, how is that situation handled?

Did you make sure that the generated network architecture is of a similar size to those baselines? I didn't find details regarding this point in the paper.

Why are the results in Figure 2 not matched to those in table 1? Agent² seems to achieve below 3500 in PPO-ant, but in table 1 it is higher than that. I may be wrong in understanding the figure or table.

---

### Official Review · Reviewer_ZYPR · 2025-10-31

**Soundness:** 2
**Presentation:** 2
**Contribution:** 2
**Rating:** 2
**Confidence:** 3

**Summary:**

This paper proposed a method named $\text{Agent}^2$, an LLM-driven 'agent-generates-agent' framework, which aims to achieve full automation of Reinforcement Learning (RL) agent design. As the authors do not provide their codes, I highly doubt its veracity because the authors did not provide the code

**Strengths:**

- This paper is easy to follow
- This algorithm is still quite creative

**Weaknesses:**

- The framework heavily relies on the code synthesis and reasoning capabilities of the underlying foundation model (LLM), and the inherent limitations of the LLM directly impact the framework's reliability
- The closed-loop iterative verification and improvement process introduces very high computational complexity
- As the code was not provided, the veracity of the experimental results remains questionable

**Questions:**

N/A

---

### Meta-Review · Area_Chair_QXL6 · 2025-12-11

**Summary:**

The reviewers raised several concerns around the experimental methodology and lack of reproducibility. Several pointed out the lack of sufficient experimental details and lack of provided code (or promises to release it). Other concerns included not using multiple random seeds (which is essential in RL experiments), inconsistencies between reported results and questions about notation. There were also questions about comparisons to baselines, for example, whether the same neural network architecture was used in the proposed method and the baselines.

All reviewers recommended rejection. Given that the authors did not submit a rebuttal, I believe it's unlikely that the reviewers would have changed their score. Therefore, I will go with the reviewer consensus and recommend rejection.

**Reviewer Concerns:**

The authors did not provide a rebuttal. Therefore, I believe all the reviewer concerns remain.

**Reviewer Scores:**

The initial reviewer scores were: 2, 2, 4, 4. Given the lack of an author response, I don't think any reviewer would have changed their score.

---

### Decision · Program_Chairs · 2026-01-26

Reject